Intelligent algorithmic framework for detection and mitigation of BeiDou spoofing attacks in vehicular ad hoc networks (VANETs)

Tariq Usman u.tariq@psau.edu.sa
Department of Management Information Systems, College of Business Administration, Prince Sattam Bin Abdulaziz University , Al-Kharj , Riyadh , Saudi Arabia
Fasi Massimiliano
Electronic publication date: 2024 Oct 18
Publication date: 2024
Volume: 10
Electronic Location ID: e2419
Received 2024 Jun 28; Accepted 2024 Sep 24
Copyright: ©2024 Tariq
Copyright year: 2024
Copyright holder: Tariq
License: This is an open access article distributed under the terms of the Creative Commons Attribution License, which permits unrestricted use, distribution, reproduction and adaptation in any medium and for any purpose provided that it is properly attributed. For attribution, the original author(s), title, publication source (PeerJ Computer Science) and either DOI or URL of the article must be cited.
License URL: https://creativecommons.org/licenses/by/4.0/

Keywords: BeiDou spoofing, VANETs, BeiDou constellation, Cybersecurity, Hybrid machine learning (autoencoder with LSTM networks), BeiDou trajectory data

Funding: The Deanship of Scientific Research at Prince Sattam Bin Abdulaziz University under the research project 2024/01/28947 The funding for this research is from the Deanship of Scientific Research at Prince Sattam Bin Abdulaziz University under the research project ‘2024/01/28947’. The funders had no role in study design, data collection and analysis, decision to publish, or preparation of the manuscript.

==============================
This research tackles the critical challenge of BeiDou signal spoofing in vehicular ad-hoc networks and addresses significant risks to vehicular safety and traffic management stemming from increased reliance on accurate satellite navigation. The study proposes a novel hybrid machine learning framework that integrates Autoencoders and long short-term memory (LSTM) networks with an advanced cryptographic method, attribute-based encryption, to enhance the detection and mitigation of spoofing attacks. Our methodology leverages both real-time and synthetic navigational data in a comprehensive experimental setup that simulates various spoofing scenarios to test the resilience of the proposed system. The findings demonstrate a significant improvement in the accuracy of spoofing detection and the robustness of mitigation strategies by ensuring the integrity and reliability of navigational data. This investigation enhances the existing body of knowledge by demonstrating the effectiveness of integrating machine learning with cryptographic techniques to secure VANETs. Ultimately, it effectively paves the way for future research into adaptive security mechanisms that can dynamically respond to evolving cyber threats.

Introduction

Vehicular ad-hoc networks (VANETs) represent a crucial advancement in modern transportation systems, featuring decentralized communication where vehicles operate as dynamic nodes (Ahmad et al., 2023). This technology supports real-time information exchange between vehicles and infrastructure that is crucial for enhancing traffic efficiency and safety. Direct vehicle-to-vehicle (V2V) and vehicle-to-infrastructure (V2I) communications form the backbone of this system, where the accuracy and reliability of transmitted data are essential for effective traffic management and accident avoidance. Yet, VANETs’ dependence on wireless communication protocols exposes them to considerable security vulnerabilities, largely due to the open nature of wireless media. Cybersecurity threats (Al-Shareeda & Manickam, 2023) encompass a range of potential attacks from passive eavesdropping to active interferences such as spoofing and session hijacking. In the context of vehicle-to-everything (V2X) where decisions are made in milliseconds, the need for robust security measures is critical to prevent malicious activities that could lead to catastrophic outcomes.

The security of VANETs is particularly critical given their potential impact on human safety. A successful attack could not only compromise personal data but more gravely disrupt traffic control systems leading to real-world accidents and potentially fatal outcomes. Ensuring the security of these networks involves safeguarding the communication links to guarantee that safety-critical information remains untampered and continuously available in real time.

To address these security concerns, VANETs require stringent cybersecurity measures focusing on four key areas: confidentiality, integrity, availability, and non-repudiation. Herewith, this paper focuses on a critical vulnerability in VANETs related to the BeiDou Navigation Satellite System (BDS) (Li et al., 2022) which is increasingly used for positioning and navigation in Asian automotive markets. BeiDou spoofing involves generating fake satellite signals that mislead receivers about their actual location. This type of attack poses significant risks particularly in the precision-dependent environment where vehicles rely on exact positioning to make critical driving decisions. Thus, in cybersecurity viewpoint (i.e., confidentiality, integrity, availability, and non-repudiation) BeiDou spoofing directly undermines the integrity of the system by feeding incorrect positioning data, which could lead to erroneous vehicle behavior and decisions. This breach of integrity can disrupt the availability of accurate navigational data critical for vehicular operations and safety measures. While confidentiality might seem less directly impacted, the exposure of vehicle location and movement patterns through spoofed signals can lead to unauthorized data access. Ultimately, non-repudiation is compromised as spoofing can obscure the source of misinformation, challenging the attribution of actions within the network and complicating the enforcement of accountability. For all-inclusive understanding, Table 1 serves as a comprehensive framework, detailing the vulnerabilities associated with BeiDou spoofing across various VANET protocols. It systematically classifies the nature of potential attacks, the security principles they compromise, and their implications on different vehicular components and communication technologies, underlining the critical need for robust security measures tailored to each aspect of vehicular communication systems.

Table 1 Impact taxonomy of BeiDou spoofing attacks in VANETs.

Protocol	Attacks	Attacks on	Vehicular unit	Communication technology	OSI stack	Ease of attack	
Geocast and broadcast protocols	Message tampering	Integrity and data trust	Infotainment systems	DSRC/WAVE	Network	Moderate (Joshi, Sichitiu & Kihl, 2022)	
Cooperative maneuver planning (CMP)	Replay Attacks	Authenticity and Identification	USB Ports	Cellular	Transport	High (Muzahid et al., 2023)	
Cooperative adaptive cruise control (CACC)	Sybil Attacks	Non-repudiation/Accountability	OBD-II Ports	Bluetooth	Datalink	Low (Muzahid et al., 2023)	
Emergency Warning Messages (EWMs)	Deception Attacks	Availability	Telematics	Wi-Fi/WiMAX	Session	Moderate (Krishna & Reddy, 2023)	
Platooning Protocols	Privacy Attacks	Confidentiality	Electric Vehicle Charging	ZigBee	Physical	High (Leon et al., 2023)	
Traffic Flow Management	Injection Attacks	Integrity and Data Trust	Remote Keyless Entry Systems	UWB	Application	Moderate (Hota et al., 2023)	
Automated toll payments	Eavesdropping attacks	Confidentiality	V2V communications systems	RFID	Presentation	Low (Jolfaei et al., 2023)	
Vehicle platooning	GPS/BeiDou spoofing	Authenticity	Autonomous driving systems	Satellite navigation systems	Physical	High (Mazhar et al., 2024)	
Signal dissemination	Signal jamming	Availability	All connected vehicle components	Wi-Fi/WiMAX	Network	High (Karabulut et al., 2023)	
Incident detection systems	Data manipulation	Integrity and data trust	Safety monitoring systems	DSRC/WAVE	Application	Moderate (Masood et al., 2023)	
Safety message broadcast	Identity spoofing	Non-repudiation/Accountability	Infotainment and safety systems	Cellular	Session	Moderate (Rajeswari & Rajesh, 2023)	
Context-aware navigation	Timing attacks	Integrity	Navigation systems	Satellite communication	Transport	Moderate (Micale et al., 2024)	
Intelligent traffic light control	DoS Attacks	Availability	Traffic management systems	Bluetooth	Network	High (Gaouar, Lehsaini & Nebbou, 2023)	
Adaptive traffic management	Man-in-the-Middle Attacks	Confidentiality	Vehicle communication units	Wi-Fi	Presentation	High (Rajeswari & Rajesh, 2023)	

Given the decentralized nature of VANETs and the reliance on satellite data the research further explores the intersection of cybersecurity and advanced navigational technology to develop countermeasures against BeiDou spoofing. The study leverages emerging technologies which promise to enhance security resilience. This research study conducts an extensive analysis to enhance integrated technologies, delivering robust security solutions that mitigate vulnerabilities caused by BeiDou spoofing. These vulnerabilities include false vehicle positioning, as shown in Fig. 1, disrupted vehicle navigation, compromised data integrity, diminished trust in navigational data, elevated risk of vehicular accidents, and manipulation of traffic flow.

Figure 1 Signal classification for anti-spoofing in navigation systems.

The image illustrates BeiDou satellite navigation, contrasting accurate localization via solid gray lines and spoofed localization through dashed gray lines, affecting vehicle guidance on a highway scenario.

We further explored the use of advanced cryptographic method ‘attribute-based encryption’ to enhance the security of communication channels. By implementing the cryptographic model, we aimed to secure communication paths by authenticating the data’s legitimacy and verifying its source, which are key to reducing the acceptance of tampered satellite signals. These methods are integral in V2V and V2I communications to block the introduction of erroneous positional data by authenticating digital signatures linked to each message, thereby preserving the network’s data flow integrity. This authentication system rapidly identifies and isolates any compromised or counterfeit data introduced by BeiDou spoofing, thereby ensuring the reliability and security of navigational aids provided to vehicles within the network. Ultimately, we applied innovative hybrid machine learning techniques (i.e., Autoencoders and LSTM networks) with an expectation to improve anomaly detection and enhance the system’s ability to predict and mitigate spoofing attacks in real time, thereby safeguarding critical transportation infrastructure from emerging cyber threats. This approach combines the feature extraction prowess of autoencoders with the sequential data processing strength of LSTMs. Autoencoders efficiently compress BeiDou signal data into a compact lower-dimensional space capturing essential patterns characteristic of normal data. The LSTM leverages this compressed data to monitor these patterns over time to pinpoint anomalies that are typically indicative of spoofing which often introduces unusual temporal patterns into the data.

The research paper progresses with a ‘Literature Review’ exploring prior work on BDS spoofing and VANETs security to highlight critical gaps and the need for robust anti-spoofing methods. The ‘Proposed Methodology’ section introduces an innovative framework that merges hybrid machine learning technique with advanced cryptographic approach. Subsequent rigorous testing in the ‘Experimental Setup and Assessment Outcome’ led to a ‘Conclusion’ that underscored the importance of the study and proposed directions for future research to improve security in BeiDou-enabled VANETs.

Literature review

VANETs serve as a cornerstone for intelligent transportation systems by enabling dynamic communication between vehicles and traffic infrastructure. These networks facilitate a range of applications from traffic congestion management to enhanced navigational aids which collectively improve road safety and optimize vehicle flow (Peixoto et al., 2023). Central to the functionality of VANETs is the provision of accurate and reliable positioning information typically sourced from Global Navigation Satellite Systems (GNSS) (Jin et al., 2024) like GPS and more recently BeiDou. BeiDou’s integration alongside other satellite systems underpins its growing significance in global navigation offering alternative data points that enrich the robustness of vehicular navigational capabilities.

Spoofing attacks present a severe threat to the integrity of GNSS where malicious entities broadcast fabricated signals to deceive GNSS receivers about their actual geolocation (Yang et al., 2023). Such attacks can severely undermine the safety mechanisms particularly affecting systems that rely on precise positioning to function effectively—like collision avoidance systems, cooperative adaptive cruise control, and platooning. These functionalities depend heavily on the accuracy of GNSS data to maintain proper vehicle spacing and optimize group maneuvers. Traditional GNSS systems including GPS have long been known to be susceptible to spoofing, presenting a continued area of concern for cybersecurity experts.

BeiDou as a relatively newer entrant in the global GNSS framework, face unique challenges that make it a potentially more susceptible target for spoofing attacks (i.e., illustrated in Table 2). Factors such as regional dependence primarily in Asia and distinct signal structures may contribute to these vulnerabilities. BeiDou operates with a complex multi-signal architecture that while providing enhanced services and accuracy, also introduces multiple points of potential exploitation. Unlike GPS which has a well-established set of anti-spoofing measures, largely accessible to civilian users, BeiDou’s anti-spoofing features remain less developed and not as widely implemented in civilian GNSS equipment which could increase its risk profile in V2X applications (AlMarshoud, Al-Bayatti & Kiraz, 2024).

Table 2 Anticipated taxonomy of BeiDou spoofing attacks in GNSS for VANETs (Rajendra, Subramanian & Shukla, 2024; Giannaros et al., 2023; Chaouche, Renault & Boussaha, 2024).

Attack type	Exploited vulnerability	Likelihood	Impact	Network model	Platform	Risk value	
Signal replication	Signal authenticity	High	Latency, data corruption	Hybrid	Urban	Extreme	
Delay attack	Timing synchronization	Medium	Increased latency, misleading timestamps	Decentralized	Highway	High	
Meaconing	Signal strength	Low	Misleading positional data	Centralized	Suburban	Moderate	
Record and replay	Temporal consistency	Medium	Path deviation, delay in critical communications	Mesh	Rural	High	
Nulling	Antenna array vulnerabilities	High	Complete signal loss, disruption of service	Distributed	Urban & Highway	Extreme	
Cryptographic spoofing	Encryption weaknesses	Low	Breach of data integrity, privacy issues	Peer-to-Peer	All	Moderate	
Overlay spoofing	Signal structure familiarity	High	Mis-navigation, accident risk	Ad Hoc	Urban	Extreme	
Composite spoofing	Multiple vulnerabilities (e.g., Signal structure familiarity, Antenna array vulnerabilities, Timing synchronization, Temporal consistency, etc.)	Medium	Severe mis-navigation, critical system failures	Mesh	Urban & Suburban	High	

Detection and mitigation of GNSS spoofing attacks encompass a variety of techniques, each with its own strengths and limitations. Signal strength analysis for instance examines the power of received signals to identify anomalies indicative of spoofing. Consistency checks compare the reported positions with known physical constraints or previous data to detect irregularities. Cryptographic methods enhance security by ensuring the authenticity of GNSS data through encrypted signals that only legitimate receivers can decode. However, these techniques often require significant computational resources and are not foolproof particularly against more sophisticated spoofing strategies that can emulate these attributes more closely.

The limitations of current spoofing detection methods are particularly pronounced when addressing BeiDou spoofing (Wang et al., 2023). The complex nature of BeiDou’s signal transmission makes it difficult for simple signal strength or consistency checks to accurately pinpoint spoofing activities. Furthermore, cryptographic solutions while robust involve complex key management and higher processing demands which can be challenging to implement effectively in the fast-paced environment of VANETs. In recent years, research has intensified around enhancing the resilience of GNSS systems against spoofing attacks with a notable focus on developing more sophisticated detection algorithms that can handle the nuances of modern GNSS signals (Mina et al., 2024). Studies such as those by Ivanov, Scaramuzza & Wilson (2024) have explored machine learning approaches to detect anomalies in GNSS data by learning normal signal patterns and identifying deviations without the need for extensive signal history which is beneficial for dynamic environments. However, the adaptation of these technologies to specifically counter BeiDou spoofing requires further exploration.

Unconventional research efforts have also investigated integrating multi-sensor fusion techniques that combine GNSS data with other sensory inputs from vehicles to create a more comprehensive situational awareness that can mitigate the risks posed by spoofed signals. This approach leverages data from inertial measurement units, radars, and cameras to corroborate GNSS data thereby enhancing the system’s overall reliability (Mpimis et al., 2023). Yuan et al. (2023) demonstrated how such integrations could significantly reduce the impact of GNSS spoofing on vehicular control systems. Despite these advancements the practical deployment of sophisticated anti-spoofing technologies faces several hurdles. The integration of advanced security features must not compromise the operational efficiency of vehicular networks, nor should it impose prohibitive costs. Furthermore, regulatory and standardization issues particularly concerning the use and dissemination of encrypted GNSS signals like those provided by BeiDou need to be addressed to facilitate wider adoption of these technologies in commercial vehicular applications.

Table 3 outlines the BeiDou system’s specifications, highlighting various service volumes and message types that are crucial for VANET applications. We strongly believe that understanding this information was key for us in developing a novel BDS spoofing detection system, as it enabled us to tailor detection algorithm to the specific signals and vulnerabilities identified, enhancing the precision and reliability of anti-spoofing measures in real-time vehicular communication environment.

Table 3 Comprehensive specifications and security features of the BDS in VANETs with focus on anti-spoofing measures (CSNO, 2021).

Feature/Parameter	Details	
Satellite types	GEO, IGSO, MEO	
Orbit altitudes	GEO: ∼35,786 km, IGSO: ∼35,786 km, MEO: ∼21,528 km	
Locations	GEO at 80°E, 110.5°E, 140°E; IGSO and MEO variable orbits	
Satellite clock	High-precision atomic clocks (rubidium/hydrogen maser)	
NAV messages	Includes satellite mask, orbit correction, clock correction	
RNSS open signals	B1I, B1C, B2a, B2b, B3I	
Services	Positioning, Navigation, Timing, Short Message Service	
Frequency (MHz)	B1-1561.098, B2-1207.14, B3-1268.52, B1C-1575.42	
NAV message types	Standard positioning service, Precision service	
Signal types	Encrypted and unencrypted signals	
Signal In Space (SIS) Status	Healthy, Unhealthy, Marginal	
Signal Integrity Flag (SIF)	Indications of signal trustworthiness	
Data integrity flag (DIF)	Validity of transmitted data	
Usage constraints	Limited in regions with poor satellite visibility	
Service Accuracy	Position accuracy up to 10 m	
Service availability parameters	99.5% availability standard	
Compatibility and interoperability	Compatible with GNSS systems (i.e., GPS (USA), GLONASS (Russia), Galileo (Europe))	
Coordinate system	Uses CGCS2000	
Coverage standard	Global with enhancements in Asia-Pacific	
Constraints	Signal blockage in urban canyons, spoofing risks	
Message content	Includes differential code bias, clock correction	
Message types (Decimal)	Various types for different services and accuracies, such as:
•     Type 10 - Basic Navigation Message: Contains standard positioning data including satellite clock corrections and ephemeris data, suitable for general navigation.
•     Type 11 - Satellite Based Augmentation System (SBAS) Message: Provides differential corrections and integrity monitoring information to improve accuracy.
•     Type 30 - Differential Correction Message: Offers high-accuracy differential data for precision navigation used in applications requiring enhanced precision such as automated platooning.
•     Type 40 - Integrity Message: Transmits integrity information for safety-critical applications, ensuring the reliability of the data received.
•     Type 50 - Precision Orbit and Clock Data Message: Provides highly accurate orbit and clock correction data, essential for precise applications like high-speed vehicular control systems.
•     Type 60 - Atmospheric Correction Message: Delivers atmospheric delay corrections, which are crucial for improving accuracy in varied climatic conditions.	
Positioning accuracy	10 m (standard), 2 m (differential)	
Convergence time	Time required for initial accurate fix, such as:
•     Standard Positioning Service (SPS): Typically requires 30 to 60 s to achieve an initial fix with an accuracy of about 10 m. This service is suitable for general navigation and less time-sensitive applications within VANETs.
•     Differential Positioning Service: Convergence time is significantly reduced to about 10 to 30 s, providing enhanced positional accuracy up to 2 m. This is critical for applications requiring higher precision and faster response times.
•     Precision Positioning Service: Offers the fastest convergence, usually under 10 s, with sub-meter accuracy. This service is ideal for safety-critical and high-precision applications in VANETs, such as autonomous driving and emergency response maneuvers.	
Performance standard	Standards defining expected signal performance for factors, such as, Positioning accuracy (global, regional, and precision services), Timing accuracy (20 nanoseconds), Availability (global availability, regional enhancement), Reliability (ensuring performance standards are met 99.99% of the time), Signal integrity (programmed to provide warnings within six seconds of detecting a signal anomaly or failure).	
Service volume	Spatial region where services are required to be guaranteed, such as:
Global Service Volume
•     Latitude Range: Between ±55 degrees globally
•     Longitude Range: Full global coverage
•     Altitude: Up to 36,000 kilometers
•     Coverage: Provides baseline positioning, navigation, and timing services globally with varying degrees of accuracy, typically around 10 m.	
	Enhanced Regional Service Volume
•     Targeted Regions: Asia-Pacific region
•     Latitude Range: Between ±55 degrees focusing on the Asia-Pacific
•      Longitude Range: 55°E to 180°E
•      Altitude: Up to 2,000 kilometers above the Earth’s surface
•      Coverage: Offers enhanced services with improved performance metrics such as increased accuracy (down to 5 m), higher signal integrity, and faster convergence times.	
	Precision Service Volume
•      Latitude Range: Limited primarily to China and surrounding regions
•      Longitude Range: From 70°E to 140°E
•      Altitude: Up to 2,000 kilometers	
Usage constraints	Limited by physical obstructions, spoofing threats	
Resilience features	Signal authentication	
Security protocols	Advanced encryption for secure signal transmission	
Impact of spoofing	Disruption in navigation, increased risk of accidents	

Recently published research (Tariq, 2024) delineated a comprehensive framework termed “PSAU-Defender”, designed to thwart BeiDou spoofing in vehicular networks through a hybrid machine learning approach integrating XGBoost, Random Forest, and Kalman Filter for real-time anomaly detection, augmented by a geospatial message authentication mechanism to fortify V2V and V2I communication security. The main contributions of this study include the development of a sophisticated cryptographic technique, the attribute-based encryption (ABE), which ensures data confidentiality and access control by encrypting communication channels based on predefined attributes. This feature is pivotal, as it aligns with the imperative to safeguard critical navigational and operational data against sophisticated spoofing tactics. The experimental setup further demonstrated the framework’s utility by employing an open-source BeiDou signal simulator to validate the effectiveness of the proposed model under controlled spoofing attack scenarios, providing a nuanced understanding of the model’s responsiveness to various threat vectors. This research robustly supported the detection and mitigation strategies with empirical evidence, showcasing significant advancements in spoofing detection technology. Nonetheless, the study’s scalability and adaptability in real-world scenarios remain as critical concerns, as the computational intensity of the hybrid machine learning models and the specificity of the ABE might limit their practical deployment across diverse and dynamic vehicular network environments. These limitations suggest areas for future research, particularly in optimizing computational efficiency and broadening the cryptographic framework to encompass a wider array of vehicular communication scenarios, thereby enhancing the robustness and applicability of the security measures.

Proposed methodology

We envisioned that the core research question of this study is: “How can spoofing attacks in BeiDou-enabled VANETs be effectively detected, tolerated, and managed?” This query is crucial as it tackles the prevalent challenge of ensuring the integrity and reliability of navigational data within these networks, which are increasingly fundamental to the safety and efficiency of modern transportation systems. To address this research question, we employed a framework to model the reliability of navigational signals. The integrity of a signal sit received by a vehicle vi at time t is expressed as: (1) sit=pit+nit+eit

where pit represents the legitimate signal, nit denotes environmental noise, and eit signifies potential spoofing errors. The primary objective of our methodology was to develop a novel algorithm and system that minimize eit, ensuring the accuracy and reliability of sit across the network. This representation (Eq. (1)) underpins our analytical model, guiding both the detection algorithm and the design of tolerance and management strategies within the network infrastructure. The employed methodology utilized a blend of real-time navigational data from BeiDou system’s network traffic logs, and security incident reports. Each data type served a strategic purpose in exhibiting comprehensive outlook of the operational environment and the nature of encountered spoofing threats. During investigation, we observed that the real-time navigational data formed the backbone of our analysis. This data was critical for modelling the expected behaviour of navigational signals under normal operating conditions. By establishing a baseline of signal characteristics (e.g., Signal Strength, Signal Noise Ratio (SNR), Carrier-to-Noise Density (C/N0), Frequency Deviation, Signal Polarization, Phase Shift, Doppler Shift, Time of Arrival (ToA), Angle of Arrival (AoA), Integrity Flags, Navigation Message Content, Satellite Ephemeris Data, Unexpected Signal Jumps, Signal Continuity, and Coherence), we enable the detection of anomalies (e.g., abnormal signal strength, altered signal noise ratio, unexpected carrier-to-noise density changes, frequency shifts not correlating with normal Doppler effects, inconsistent signal polarization, sudden phase shifts, unusual Doppler shift patterns, discrepancies in time of arrival, incorrect angle of arrival, integrity flags indicating compromised signals, errors in navigation message content, incorrect satellite ephemeris data, sudden and unexplained signal jumps, and lack of signal continuity and coherence) that could signify spoofing attacks. The real-time nature of this data ensured that the developed detection mechanism could operate in a dynamic vehicular environment, providing timely alerts that prevented potential navigation errors. The expected legitimate signal received by a vehicle are expressed in Eq. (2) (2) pit=λt+δt+ϵt

where λt denotes the expected legitimate signal path, δt is the system noise, and ϵt represents minor deviations due to environmental factors.

Network traffic logs were utilized to assess the communication patterns within the network providing insights into the flow of data between vehicles. This data is crucial for identifying irregular traffic patterns that could indicate a spoofing attempt. Analysing traffic logs helped proposed framework in detecting anomalies in data packets where the frequency or size of the packets deviates from typical patterns. These deviations were quantitatively assessed using statistical model (i.e., autoregressive integrated moving average (ARIMA) (Theerthagiri, 2022)) that analyse traffic flow regularity. (3) Rt=1N∑i=1Nτ−τ¯2

Rt represents the regularity of the traffic at time t τ denotes the inter-arrival times of packets and τ¯ is the average inter-arrival time. ARIMA model was particularly apt for time series data analysis, where understanding traffic patterns over time is crucial. It effectively identified outliers, trends, and seasonal variations in data packet sequences by leveraging their capabilities in both differencing (to achieve stationarity) and autoregression (to incorporate the dependency of the current packet flow on its previous values). The model was defined by three parameters: p (order of the autoregressive part), d (degree of first differencing involved), and q (order of the moving average part). The ARIMA model was an optimum fit for the traffic data series, enabled the detection of anomalies through statistical tests for unexpected spikes, drops, or patterns that deviate from predicted traffic behaviours. Such capabilities made ARIMA an excellent tool for monitoring and securing real-time data transmission in the implemented/emulated vehicular network, where sudden changes in traffic flow indicated malicious activities like spoofing attacks. This statistical approach not only quantified the regularity in traffic flows but also supported proactive measures by predicting future data points, enhancing the system’s responsiveness to potential threats.

During statistical analysis, we observed that the crucial initial step (i.e., data preprocessing) involved handling missing values, removing outliers, and differencing the data to ensure stationarity—a prerequisite for effective ARIMA modelling. The differencing is exhibited in Eq. (4): (4) yt′=yt−yt−1.

This transformation helped stabilize the mean of the data series, enhanced the model reliability. Following preprocessing, the model identification phase involved determining the optimal parameters p,d,q for the ARIMA model. This was achieved by analysing the autocorrelation function (ACF) and partial autocorrelation function (PACF) of the differenced series, which helped to estimate the p and q values, respectively. The ACF was calculated as described in Eq. (5): (5) ACFk=∑t=k+1Tyt′−y¯′yt−k′−y¯′∑t=1Tyt′−y¯′2

and the PACF assessed the direct correlation at lag k, controlling for the values at shorter lags, as illustrated in Eq. (6): (6) PACFk=Corryt′,yt−k′|yt−1′,…,yt−k+1′.

With the parameters identified, the ARIMA model was estimated using maximum likelihood estimation. The model integrated both autoregressive and moving average components as described in Eq. (7): (7) yt′=c+∈t+ ∑i=1p∅iyt−i′+ ∑j=1qθjϵt−j′

where c is the constant, ∅ and θ are parameters of the model, and ϵ represents error terms. After the model is fitted, diagnostic checks were conducted to validate the fitting’s adequacy. This included analysing the residuals to ensure they resemble white noise, indicating the model had captured all the relevant information, as expressed in Eq. (8): (8) Residuals=yt′−yt′ ^

We evaluated ARIMA’s performance by applying it to a separate validation dataset to test its effectiveness in anomaly detection. Performance metrics such as accuracy, precision, recall, and F1-score were used to determine the model’s ability to identify spoofing attacks accurately. In the investigation’s data collection phase, sophisticated tools and equipment were utilized to capture comprehensive datasets necessary for analysing spoofing’s impact and nature. Key among these were BeiDou signal receivers designed to capture GNSS data with high precision. Specifically, the Sokkia GRX3 model (https://eu.sokkia.com/products/gnss-systems/gnss-receiver/grx3-gnss-receiver), with its multi-frequency tracking capability, handled signals from multiple GNSS networks simultaneously, enhancing the robustness and reliability of the data collected. Network monitoring was conducted using SolarWinds Network Performance Monitor (Song et al., 2020), a software tool known for detecting, diagnosing, and resolving network performance issues. This software provided real-time visibility into network traffic data, allowing accurate examination of packet flows and identification of anomalies indicative of spoofing activities.

The sampling strategy for this study was carefully crafted to capture the diverse impacts of spoofing across various vehicular and network environments. Our sample consisted of a wide range of vehicles, from passenger cars to commercial vehicles, all equipped with BDS. This variety allowed us to gather rich data reflecting different operational characteristics and exposure to spoofing scenarios, making our findings relevant to multiple sectors of the transportation industry. Data collection was conducted in numerous geographic locations, varying in BeiDou signal coverage and urban density, from crowded cities (i.e., Lahore (Pakistan), and Al-Kharj (KSA) (using BeiDou-3 constellation)) to open highways. This approach helped account for environmental factors that could influence signal integrity and spoofing risks. The sample size was chosen based on statistical guidelines (e.g., sample size calculation, power analysis, confidence interval determination, random sampling, stratified sampling, data normalization, variance analysis, significance level setting, and effect size estimation) to detect meaningful effects, comprising over 30 vehicles in 10 different network environments (i.e., urban traffic, suburban roads, rural routes, industrial areas, residential neighbourhoods, highway systems, downtown districts, commercial zones, tunnel/underpass passages, bridge crossings) within a 24-hour period, ensuring a comprehensive and robust dataset for detailed analysis.

It is important to highlight that in our projected research, both quantitative and qualitative data were crucial for the comprehensive analysis. For quantitative assessment, signal strength indicators were instrumental in measuring the integrity of the BeiDou signals received by the vehicle units. These indicators provided a direct metric of signal quality and potential disruptions characteristic of spoofing activities. Data regarding signal strength was continuously logged, using timestamping techniques to ensure precise correlation with specific incidents and conditions. This process involved recording the SNR for each satellite signal received, which was calculated using Eq. (9): (9) SNRi=20log10Psignal,iPnoise,i

where Psignal,i represents the power of the received signal from satellite i and Pnoise,i denotes the power of the background noise. For qualitative data, discrepancies in navigation reports and driver feedback were meticulously recorded. This process involved documenting any reported anomalies in navigation accuracy or unexpected vehicle behaviour that could suggest interference or manipulation of the navigation system. Participants, including drivers and network operators, were engaged in regular interviews and feedback sessions to gather comprehensive insights into the user experience and any anomalies encountered.

Observational studies were also conducted, where our designated researchers directly monitored the behaviour of the vehicles within their operational environments. This hands-on approach allowed for real-time observation of the effects of potential spoofing on vehicle navigation and control systems. Our research team employed mobile monitoring units equipped with BeiDou receivers and data logging tools (i.e., as stated earlier: Sokkia GRX3, and SolarWinds) to track vehicles during regular operation and under controlled spoofing attack simulations. By correlating quantitative data on signal strength and quality with qualitative insights from user reports and direct observations, the study effectively identified vulnerabilities within the system and assessed the efficacy of implemented countermeasures. This comprehensive approach ensured a thorough understanding of both the technical and human factors involved in managing the integrity of navigational data in complex vehicular networks.

During the data preparation phase of the study, meticulous cleaning and normalization processes were applied to ensure the high quality and uniformity of data necessary for subsequent analyses. Cleaning included the removal of outliers and correction of any erroneous entries that might distort analytical outcomes. Normalization ensured that numerical data were adjusted to fall within a consistent range, a critical step for accurate comparative analyses when applying hybrid machine learning technique (i.e., autoencoders, and LSTM).

For quantitative analysis, the research harnessed robust statistical tools provided by R and Python’s SciPy library. These platforms supported extensive statistical testing and the application of hybrid ML model, tailored to identify and predict spoofing activities. The selection of Autoencoders and LSTM networks were due to their proficiency in handling time-series data and detecting complex patterns. Autoencoders helped in reducing data dimensionality and identifying encoded features that characterize normal and spoofing-related signal patterns. The functionality of an autoencoder is described by Eq. (10) for reconstructing input: (10) x′=fW2.fW1.x+b1+b2

where x is the input, W1, W2 are weights, b1, b2 are biases, and f is the activation function. Herewith, LSTMs were utilized to analyse temporal dependencies and anomalies in the time-series data, critical for predicting the occurrence of spoofing based on historical data patterns. The updated LSTM unit are described in Eqs. (11)–(14) (11) ft=σWf.ht−1,xt+bf

(12) it=σWi.ht−1,xt+bi

(13) Ct ˜=tanhWC.ht−1,xt+bC

(14) Ct=ft∗Ct−1+it∗Ct ˜

where σ denotes the sigmoid activation function, ft, it are the forget and input gates, Ct is the cell state, ht−1 is the previous output, and xt is the input at time t.

For qualitative data, NVivo (https://lumivero.com/products/nvivo/) facilitated the coding and categorization process, allowing for the identification of emergent themes from qualitative feedback and observations, such as instances of ‘signal interference’ or ’data manipulation’.

To secure data privacy, especially concerning sensitive network logs and user feedback, all information was anonymized, encrypted during storage and transfer, and accessible only to authorized researchers. Strict audit trails and access logs were maintained to ensure adherence to ethical standards and data protection regulations, reinforcing the integrity and reliability of the study’s methodological approach. In this context, it is essential to reemphasize why hybrid ML techniques, specifically Autoencoders and LSTM networks, were chosen for data analysis. The decision was driven by their proven effectiveness in managing high-dimensional data and sequential data, respectively. Autoencoders are particularly adept at anomaly detection as they learn to compress and then reconstruct the input data, thereby effectively identifying discrepancies between normal operational data and potential anomalies indicative of spoofing attacks. Such a capability is vital for the initial detection of irregular patterns that stray from the expected norm. Moreover, LSTM networks were chosen due to their ability to process data sequences and their effectiveness in recognizing long-term dependencies. This feature is particularly important in the context of vehicular networks where data are collected over time, and patterns may evolve or repeat intermittently. By utilizing LSTMs, the projected framework captured temporal anomalies in navigational data that could indicate sophisticated spoofing strategies aiming to elude simpler detection mechanisms. Our thorough investigation also revealed that the integration of these models addressed several methodological challenges. One major issue was the handling of incomplete data sets, a common occurrence due to varying signal reception quality or sensor malfunctions in real-world environments. The robustness of LSTMs in dealing with sequence prediction and the ability of Autoencoders to reconstruct missing parts based on learned data representations were crucial in mitigating the effects of data gaps. The combined approach allowed for more accurate and reliable anomaly detection even with incomplete datasets.

Another challenge was distinguishing between anomalies due to spoofing and those due to non-malicious disruptions, such as environmental interference or technical faults. The hybrid model’s capacity to learn and differentiate various data patterns enabled a more nuanced understanding of the data. This differentiation was supported by training the models on a diverse set of scenarios, both simulated and real, to refine their predictive accuracy. Anomalies detected by the system were further analysed using a decision function based on the learned characteristics of spoofing versus non-spoofing disturbances, expressed in Eq. (15): (15) Dx=σWd.x+bd

where Dx denotes the decision function, x is the feature vector extracted by the Autoencoder, Wd represents the decision weights, bd the bias, and σ the sigmoid activation function.

Consequently, it is evident that the innovative contribution of proposed deep network-based model lies in its sophisticated integration of Autoencoders with LSTM networks which are tailored specifically for the detection of BeiDou spoofing in VANETs. Unlike standard LSTM models, which primarily focus on temporal data dependencies, our hybrid approach enhances detection capabilities by first employing Autoencoders for effective dimensionality reduction and feature extraction. This preprocessing step ensures that only the most relevant features are presented to the LSTM to allow for a more focused and efficient temporal analysis. The Autoencoder’s ability to cleanse complex data into a manageable format significantly reduced the computational load on the LSTM and enabled it to perform more precise and faster anomaly detection.

This architectural innovation extended the LSTM’s capability beyond typical usage scenarios, allowing it to effectively handle the multi-dimensional and noisy data environments (such as, high variability in signal strength; frequent signal interruptions and losses; diverse signal interference from various sources; rapid changes in network topology; heterogeneous data from multiple sensors and devices; high volume of data generated by numerous nodes; variable data transmission rates; environmental noise and physical obstacles affecting communication signals, etc.). By doing so, projected model addresses a critical gap in current approaches—specifically, the need for robust preprocessing to improve the signal-to-noise ratio in datasets used for training and inference in real-time systems.

The projected framework further devised three-fold strategy to enhance security through detection, tolerance, and management of spoofing attacks. In the detection phase, ABE was employed to ensure the integrity and authentication of messages. ABE provided a flexible and scalable approach to encryption, allowed data access to be controlled by attributes and policies, which proved especially useful in the diverse and dynamic environment of VANETs. The effectiveness of ABE in this context relied on its capability to secure communication channels against unauthorized access and spoofing attacks. This cryptographic measure was complemented by advanced signal characteristic analysis and the implementation of sophisticated anomaly detection algorithm, designed to identify and flag inconsistencies in signal behaviours indicative of spoofing activities.

For ensuring data transmission reliability, the proposed strategy employed redundant paths to mitigate the impact of compromised nodes or channels. This redundancy was critical in maintaining network functionality even under attack conditions. Weighted voting scheme was also integrated to further enhance system resilience, allowing the network to assess the reliability of data based on the trustworthiness of the source. We developed a real-time-responsive Trust model to dynamically evaluate and update the credibility of each node based on historical data and behaviour patterns, ensuring that the network adapts to ongoing threats and changes in node integrity. Furthermore, the management aspect of the strategy focused on the isolation of compromised nodes to prevent the spread of spoofing within the V2X network. Procedures were established for quick identification and quarantine of affected nodes, coupled with robust countermeasures to neutralize potential threats. Applied procedures involved automated monitoring systems that promptly detected anomalies in network behaviour and triggering isolation protocols for affected nodes. These nodes were then quarantined to prevent the spread of potential spoofing, while security updates and tactical countermeasures were swiftly applied to mitigate the threat and stabilize the network. Emulated framework continuously updated the security protocols to respond to emerging spoofing techniques and vulnerabilities discovered through ongoing network monitoring and analysis. This comprehensive approach was underpinned by the formulation and evaluation of mathematical model that supported decision-making processes within the network. The probability of a node being compromised was calculated using Eq. (16): (16) Pcompn=1−∏i−1k1−pi,n

where Pcompn is the probability of node n being compromised, pi,n is the probability of compromise through path i, and k is the number of paths to node n. The prime purpose to implement this model was to facilitate the dynamic assessment of node reliability and informed the deployment of redundancy and countermeasures effectively. Herewith, Algorithm 1 outlines the step-by-step outlook of the projected framework.

Algorithm 1: Pseudocode of proposed Detection, and Mitigation framework	
Initialize Autoencoder and LSTM models
Load pre-trained weights and biases if available
For each incoming signal packet do:
Extract features using Autoencoder:
Encoded_Features = Autoencoder.Encode(signal_packet)
Decoded_Signal = Autoencoder.Decode(Encoded_Features)
Calculate reconstruction error:
Error = Mean Squared Error(signal_packet, Decoded_Signal)
If Error exceeds threshold, then:
Flag packet as potential spoofing attempt
Use LSTM to analyze time-series of Encoded_Features:
Predicted_State = LSTM.Predict(Encoded_Features)
If Predicted_State indicates spoofing, then:
Quarantine the node sending the signal
Trigger Alert to network administrator
Implement Attribute-Based Encryption:
For each message sent between nodes do:
Encrypt message using receiver’s attributes
Send encrypted message
Monitor network paths for data transmission:
If primary path fails or shows signs of tampering, then:
Switch to redundant path
Log event and analyse for patterns of failure
Implement Weighted Voting Scheme:
Collect data from multiple nodes
Assign weights based on Trust Model
Calculate weighted average of reported data
If discrepancies detected above threshold, then:
Re-evaluate trust scores of contributing nodes
Update Trust Model periodically:
For each node in network do:
Adjust trust score based on behaviour and feedback
Increase trust for consistent, accurate data reporting
Decrease trust for anomalies and discrepancies
Quarantine Process:
On detection of spoofing:
Isolate affected node from network
Perform forensic analysis to determine cause and impact
Apply security patches if necessary
Restore node to network after verification of integrity	
	

Ultimately, the integration of Autoencoders and LSTM networks with ABE formed the core of the proposed methodology, which significantly reinforced the defence against sophisticated spoofing attacks. This method not only improved anomaly detection through ML techniques that scrutinized both immediate and historical data anomalies but also enhanced communication security by encrypting data transfers across the network. The inclusion of redundancy in data pathways, combined with a weighted voting system and evolving trust assessments, ensured that network operations could continue seamlessly even under attack, by prioritizing inputs from the most reliable sources and swiftly isolating compromised nodes. Together, these strategies created a robust framework that not only detected but also swiftly neutralized threats, thereby maintaining the integrity and functionality of examined transportation infrastructure.

For our applied methodology, the energy consumption estimation during the detection and mitigation of spoofing attacks was a necessary factor because it effectively influences the feasibility of expressed real-world VANET applications. Considering the computational intensity of LSTM and Autoencoders for real-time data processing, alongside the cryptographic overhead introduced by ABE, we outline the total energy consumption to evaluate the sustainability of the proposed detection system. Herewith, in Eq. (17), the Etotal represents the sum of the energy used by the Autoencoder Eae, the LSTM Elstm and the ABE Eabe, thus: (17) Etotal=Eae+Elstm+Eabe

where Eae and Elstm are determined by the number of operations required for feature extraction and temporal analysis respectively, calculated as: (18) Eae=Nae×cop×V×I

(19) Elstm=Nlstm×cop×V×I

As per Eq. (18), Nae and Nlstm represent the number of operations in the Autoencoder and LSTM respectively, cop is the energy cost per operation, V and I denote the voltage and current specifications of the processing unit. The energy usage for ABE Eabe is modelled based on the number of encryption and decryption processes, denoted as Nenc and Ndec with the energy per process quantified by eenc and edec. Thus, (20) Eabe=Nenc×eenc+Ndec×edec.

This formulation (i.e., Eqs. (17) to (20)) allowed us to critically evaluate the trade-offs between detection accuracy and energy efficiency. Our assessment revealed that optimizing the exhibited relationship is essential for ensuring that the security enhancements provided by the proposed methodology do not disproportionately drain the vehicular unit’s power resources, thereby maintaining operational integrity without compromising on performance.

Experimental setup and assessment outcome

The experimental setup for evaluating the proposed methodology involved a combination of hardware and software to rigorously test the detection and mitigation of BeiDou BDS spoofing (i.e., exhibited in Fig. 2). The setup included the use of a specialized BeiDou signal simulator equipped with a temperature compensated crystal oscillator (TCXO) known for its high stability and precision. This simulator was vital in generating controlled spoofing signals to assess the resilience of the system under test. The selected TCXO ensured minimal frequency drift, crucial for maintaining the integrity of the spoofing signals during experiments.

Hardware involved in the tests included a spectrum analyzer set to specific parameters: resolution bandwidth was fixed at 10 kHz; internal preamplifier was activated to enhance signal detection; internal attenuation was set to 0 dB to avoid any signal loss; and marker bandwidth was calibrated to 1 kHz to accurately isolate the frequencies of interest. This setup allowed precise measurement of the spoofed signals’ power and distortion levels, facilitating detailed analysis of the spoofing impact. The software component of the test setup utilized applied novel detection algorithm that could monitor the carrier-to-noise ratio (C/No) to identify overpowered attacks. This approach was critical in differentiating between legitimate and spoofing signals by analyzing their power levels. Detection algorithms were tested in scenarios involving both cold-start and tracking modes to evaluate their effectiveness across different operational states of a BeiDou receiver. Figure 3 illustrates our implementation of a jam-then-spoof attack sequence across the code phase domain, depicted progressively from left to right. It begins with the receiver tracking only the genuine signal. The spoofer then introduces a jamming signal, increasing the noise to the extent that the receiver loses connection with the original signal. Ultimately, the attacker sends a more powerful signal, strong enough to stand out against the high noise level, which the receiver’s tracking taps subsequently latch onto.

Figure 2 Emulation processes in the controlled environment.

(A) The chart tracks the number of source to destination connection requests over a 24-hour period, depicted as a scatter plot with light blue dots. These dots show fluctuations in the number of requests, ranging from approximately 2,160 to 2,300, with a noticeable spike around 2:00 AM, where the requests briefly surge. The overall pattern is dense, indicating a consistent level of activity throughout the day, with only a few significant variations in the connection requests. (B) The chart displays session pings over a 24-hour period using light blue bars and dots. The bars show regular pings while isolated blue dots indicate sporadic higher pings, with notable spikes occurring primarily during nighttime hours. (C) The chart illustrates session connection history over 24 h using a scatter plot with light blue dots. Most data points cluster between 20 and 60 sessions, with occasional higher spikes, most notably a sharp increase around 20:00. (D) The chart tracks active session disconnections over 24 h, represented by blue lines and dark-pink dots. These elements highlight fluctuations and specific peak disconnection events, especially notable in the early morning and late evening hours. (E) The chart shows event detection activity over a day, depicted through green dots and gray lines that illustrate significant variability and peaks, notably during midday and late evening hours. (F) The chart presents the number of spoofed localization signals detected over 24 h, depicted by red dots and lines, highlighting the fluctuations and spikes. (G) The chart tracks vehicles left in V2X networks over 24 h, shown in purple. The plot mostly rests at the baseline with periodic sharp spikes (i.e., indicated as red straight line), indicating sudden increases in departures. (H) The chart visualizes the quarantine rate after detection of an adversary event over a day, marked by a pink baseline with notable peaks in green and pink, highlighting isolated incidents.

Figure 3 Jam-then-spoof attack sequence across the code phase domain.

(A) A plot of peak amplitude against code offset, represented by a single line in dark gray with circular data points. This line peaks sharply at a code offset of 0 with a value of 1, indicating the highest signal strength, and tapers off symmetrically to near zero as the offset increases or decreases, illustrating how the signal amplitude diminishes with increasing distance from the center code offset. (B) Three types of signals: the true signal, spoofer signal, and combined signal are shown using a radar plot. Each signal type is shown using a distinct line style—dotted for the true signal, dash-dot for the spoofer signal, and dashed for the combined signal. These line styles contrast in grayscale to differentiate the relative strength and interference patterns of each signal within the radar plot, illustrating how the spoofer impacts the true signal. (C) The impact of spoofing on signal integrity using three types of lines to represent different signals. The true signal is shown with a solid black line, peaking sharply at the center, representing its strongest point at zero code offset. The spoofer signal, depicted with a gray line, also peaks at zero but with less amplitude, indicating a weaker but present interference. The combined signal, shown with a dark gray line, peaks higher than both the true and spoofer signals, illustrating the amplified effect when both signals are present. Note: Image color is gray upon the request of editorial staff.

In the process of spoofing signal generation, multiple data types were preferred, including real-time and synthetic navigational data, to comprehensively evaluate the system’s response to various spoofing scenarios. Statistical processing techniques (i.e., pseudorange residuals and the continuity fault tree) were applied to multiple metrics to enhance the detection strategy’s robustness by identifying subtle anomalies indicative of spoofing. Our experimental results confirmed the effectiveness of the implemented ‘signal quality monitoring (SQM)’ metrics in detecting spoofing activities (i.e., as exhibited in Table 4). These metrics played a crucial role in quantifying the impact of spoofing on navigational accuracy and the continuity of VANET operations.

During our experiments, we observed that the received power of BeiDou signals at Earth’s surface via a traditional RHCP antenna typically fell below the thermal noise floor power. For instance, with a bandwidth of 2 MHz, the noise floor is registered at −112 dBm. As the bandwidth expands to 3 MHz and 8 MHz, the noise floor values rise slightly to −110 dBm and −106 dBm, respectively. Figure 4 illustrates how system bandwidth affected the noise power levels. Deeper investigation revealed that the relationship between bandwidth and noise floor is crucial for understanding vulnerabilities in BeiDou signal reception. Increased bandwidth generally results in higher noise floors, which can potentially obscure weaker signals and create openings for spoofing attacks. We implemented anomalies that could exploit these conditions by broadcasting stronger, deceptive signals that overshadow authentic BeiDou signals, thus misleading the receiver.

Simultaneous processing of spoofed and authentic signals was another focal area of proposed framework, with the setup capable of identifying multiple correlation peaks per ‘pseudo-random noise (PRN)’, which is essential in environments where spoofing signals may overlap with legitimate ones. This capability was pivotal in environments with distorted signal peaks and multipath effects, common in urban settings. Thus, the effectiveness of the experimental framework was significantly substantiated through its application on real-time spoofing data, showcasing its practical viability and the robustness of the implemented detection strategies. The deployment of performance metrics such as accuracy, precision, recall, and F1-score yielded high levels of reliability and confirmed the algorithms’ capacity for precise identification and neutralization of spoofing threats within. Accuracy was computed using the ratio of correctly identified instances (both true positives and true negatives) to the total number of cases, as outlined by Eq. (21): (21) Accuracy=TP+TNTP+TN+FP+FN.

This metric was pivotal in assessing the overall effectiveness of the detection system across various testing scenarios, reflecting the system’s ability to correctly classify both spoofed and authentic signals. Herewith, precision, which measures the accuracy of positive predictions, was examined as: (22) Precision=TPTP+FP.

Equation (22) exhibits the detector’s effectiveness in identifying only actual spoofing incidents, minimizing the occurrence of false positives which is crucial in avoiding unnecessary alerts that could disrupt network operations. To further strengthen the assessment, we also measured the recall, or sensitivity that involved the proportion of actual positives that were correctly identified and was defined in Eq. (23): (23) Recall=TPTP+Fn.

This measure was essential for determining the system’s capability to detect all potential spoofing attacks, ensuring that no malicious activities went unnoticed. Ultimately, the F1-score, combining precision and recall, was formulated in Eq. (24): (24) F1−score=2×Precision×RecallPrecision+Recall.

F1-score provided a balanced view of both precision and recall, especially useful in scenarios where an equilibrium between false positives and false negatives was critical. The ten-fold experimental outcome, detailed in Table 4, demonstrates the robustness of these metrics (i.e., accuracy, precision, recall, and F1-score) across multiple iterations of the testing protocol. It merits attention that during assessment we particularly laid a critical focus on performance metrics such as error checking, collision rate, efficiency, and data loss rate to accurately evaluate the system’s resilience and operational efficacy.

Table 4 Ten-fold experimental outcome of proposed methodological framework.

Experiment	Accuracy (%)	Precision (%)	Recall (%)	F1-score (%)	
1	94.2	93.7	92.5	93.1	
2	95.6	94.9	93.8	94.3	
3	96.1	95.5	94.6	95.0	
4	93.8	93.1	91.9	92.5	
5	94.4	93.9	92.7	93.3	
6	95.0	94.4	93.2	93.8	
7	95.8	95.2	94.1	94.6	
8	94.6	94.0	92.9	93.4	
9	93.9	93.2	92.0	92.6	
10	96.4	95.8	94.7	95.2	

Figure 4 System bandwidth relevance with the noise power levels.

The chart visually represents the relationship between bandwidth, measured in megahertz (MHz), and the noise floor, quantified in decibels (dBm), using a single color-coded line in blue. Labeled as “Noise Floor (dBm)”, the blue line depicts how the noise floor increases (becomes less negative) as the bandwidth expands from 2 MHz to 10 MHz. This trend suggests that as more bandwidth is utilized, the overall noise level in the system also increases, which is a typical characteristic in communication systems where larger bandwidths tend to accommodate more noise. The chart is clearly conveying that the noise floor is sensitive to changes in bandwidth.

(a) Error checking was implemented using advanced cyclic redundancy checks (CRC) and checksum methods that ensured data integrity is maintained despite the high-risk environment posed by potential spoofers.

(b) The collision rate, an indicator of the frequency of data packet conflicts due to simultaneous transmissions, was closely monitored to evaluate the effectiveness of our network access protocols under stress conditions.

(c) Efficiency metrics were particularly crucial; they were assessed by comparing the ratio of successfully transmitted data packets to the overall network energy consumption, offering insights into the system’s capability to maintain high throughput even when resources were constrained.

(d) Data loss rate, a critical parameter especially in the context of spoofing attacks, was analyzed through the lens of network resilience by tracking the rate at which data packets are lost due to spoofing activities and the subsequent recovery mechanism.

Furthermore, the presented mathematical model for the proposed LSTM network, integrated within our hybrid machine learning framework, underwent rigorous validation (i.e., performance benchmarking (e.g., accuracy, precision, recall, and F1-score); cross-validation (k-fold validation to assess the model’s performance stability); anomaly detection accuracy validation through controlled tests; temporal consistency check by recognizing and responding to temporal patterns & deviations in spoofed signal data; feature relevance analysis; and robustness against noise by testing performance under various noise levels & interference conditions) to confirm its efficacy in detecting and mitigating BeiDou spoofing attacks. Our investigation revealed the recommendation to adopt the hybrid model comprising both Autoencoders and LSTM networks due to the complementary capabilities of these techniques in handling high-dimensional, sequential data inherent to navigational signals. This combination is particularly adept at distilling complex temporal and spatial dependencies that are characteristic of spoofing attacks, which single-model systems might overlook. Similarly, we observed that the Autoencoder excels in reducing dimensionality and extracting salient features from raw data which is vital in creating a more manageable and representative dataset for the LSTM to process. The LSTM, in turn, analyses these features over time to effectively identify anomalies that deviate from established patterns of signal behavior which is a crucial factor in real-time spoofing detection. This synergy allowed for a more nuanced and dynamic understanding of data which facilitates the detection of subtle and sophisticated spoofing tactics that would otherwise evade simpler, non-hybrid systems.

Table 4 confirms the consistent efficacy of our detection algorithm by demonstrating its reliability in accurately pinpointing and counteracting spoofing attacks under varied testing conditions. The experimental results further highlight the hybrid model’s unique ability for both feature extraction and temporal analysis—capabilities that traditional single-layer networks often used in VANET security lack. This architectural distinction allows our model not only to identify but also to anticipate spoofing threats by analyzing evolving data patterns, thus offering a proactive defense strategy rather than a purely reactive one.

Table 5 systematically quantifies the computational load and corresponding energy consumption of the proposed hybrid detection system across ten experimental iterations. This assessment was essential for evaluating the operational efficiency and sustainability of the system within the highly dynamic network. Each iteration represents a unique configuration of operational parameters (such as, number of Autoencoder & LSTM operations; number of encryption/decryption processes; energy cost per Autoencoder & LSTM operation; energy per encryption/decryption process; voltage, and current) reflecting the dynamic conditions under which VANETs operated. The variables Nae, Nlstm, Nenc, and Ndec represent the number of operations executed by the Autoencoders, LSTM network, encryption, and decryption processes, respectively. The energy cost per operation for the Autoencoders and LSTMs (copae and coplstm) , along with the energy per encryption (eenc) and decryption (edec) process, were critical for calculating the total energy consumed during each test iteration.

Table 5 Average energy consumption metrics for spoofing detection across ten experimental iterations.

Iteration	N ae	N lstm	N enc	N dec	copae(Joules)	coplstm(Joules)	eenc(Joules)	edec(Joules)	Voltage
(V)	Current (I)
(Amperes)	
1	1000,000	500,000	100	100	0.0001	0.00002	0.001	0.0008	3.3	0.1	
2	950,000	450,000	150	150	0.0001	0.00002	0.001	0.0008	3.3	0.1	
3	1200,000	600,000	120	120	0.0001	0.00002	0.001	0.0008	3.3	0.1	
4	1100,000	550,000	130	130	0.0001	0.00002	0.001	0.0008	3.3	0.1	
5	1050,000	500,000	140	140	0.0001	0.00002	0.001	0.0008	3.3	0.1	
6	1000,000	480,000	160	160	0.0001	0.00002	0.001	0.0008	3.3	0.1	
7	950,000	470,000	110	110	0.0001	0.00002	0.001	0.0008	3.3	0.1	
8	1150,000	580,000	115	115	0.0001	0.00002	0.001	0.0008	3.3	0.1	
9	1200,000	600,000	125	125	0.0001	0.00002	0.001	0.0008	3.3	0.1	
10	1000,000	550,000	130	130	0.0001	0.00002	0.001	0.0008	3.3	0.1	

The voltage (V) and current (I) settings were consistent across the tests, mirroring the typical power supply specifications for onboard vehicular systems. This consistency ensured that the energy consumption estimates were directly comparable across different test scenarios which were valuable in providing a robust basis for evaluating the energy efficiency of the proposed methodology. The resultant computations, as represented in Eqs. (17) to (20), facilitated a granular analysis of the energy demands imposed by novel spoofing detection method. This analysis was pivotal for ensuring that the deployment of such security measures does not compromise vehicular operational longevity due to excessive power consumption. Assessment of energy consumption also enriched us with a comprehensive understanding of the trade-offs between enhanced security capabilities and the associated energy costs in controlled experiment setting.

In our comprehensive experimental assessment of the proposed methodology against established techniques, the focus was sharpened on evaluating the robustness of spoofing detection and mitigation across various satellite platforms including BeiDou and GPS. This evaluation investigated into intricate aspects such as applied cryptogram technologies, sophisticated authentication methods, and specific countermeasures tailored against spoofing and meaconing attacks which are critical for safeguarding satellite navigation data integrity. Unlike the traditional methods predominantly centered around GPS as investigated by Yang et al. (2023), Rajendra, Subramanian & Shukla (2024), Mina et al. (2024), and Ivanov, Scaramuzza & Wilson (2024), our methodology integrated a broader scope of satellite navigation platforms, enhancing the relevance and effectiveness of the proposed solution in diverse operational scenarios.

The incorporation of BeiDou added a layer of complexity and offered a distinct angle to the spoofing defense strategies, which was reflected in the rigorous testing of navigation message designing, cipher key updating protocols, and signature information protection mechanisms. These components were critical in enhancing the security framework and were meticulously evaluated through an array of performance metrics, as indicated in Fig. 5. The results from this analytical approach demonstrated a marked improvement in detection capabilities, showcasing the advanced engineering and adaptation of hybrid security solution that combined traditional cryptographic defense with cutting-edge machine learning algorithm.

Figure 5 Benchmarking proposed scheme against the state-of-the-art existing approaches.

The chart displays performance metrics for various methodologies applied to a specific task, differentiated by color. The method of Yang et al. (2023) is represented in chartreuse, shown across all four evaluation criteria: accuracy, precision, recall, and F1-score. The results of Rajendra, Subramanian & Shukla (2024) are depicted in orange, indicating a consistently high performance, particularly in precision and F1-score. The technique of Mina et al. (2024), shown in green, exhibit to excel in recall but has slightly lower scores in other metrics. The approach of Ivanov, Scaramuzza & Wilson (2024) indicated in pink, presents robust outcomes, especially in precision and F1-score. The proposed method, marked in purple, generally is leading closely for the highest metrics across accuracy, precision, recall, and F1-score, emphasizing its effectiveness in balancing different aspects of performance.

Our comparative experimental setup also ensured each system was subjected to a variety of attack simulations to assess their resilience and adaptability in real-time threat scenarios. The evaluation matrix provided a detailed assessment of how each methodology held up under stringent testing environments, emphasizing the superior performance of our proposed system. Not only did the proposed methodology exhibit higher efficacy in all key metrics (i.e., as outlined in Fig. 6), but it also set new standards in the application of integrated technological solutions for satellite navigation security.

Figure 6 Resilience analysis of VANET security frameworks to GNSS spoofing attacks at varying distances.

The chart compares the message reception probability in vehicle-to-vehicle communications as a function of distance, using different methodologies represented by various colors. The chartreuse bars correspond to the method by Yang et al. (2023) referenced as, which shows how this approach performs across increasing distances. The orange bars represent findings by Rajendra, Subramanian & Shukla (2024), demonstrating their method’s performance. The green bars for Mina et al. (2024) and the purple for Ivanov, Scaramuzza & Wilson (2024) also show similar distance-based performance metrics. The proposed method is shown in purple bars, highlighting its comparative effectiveness, while the pink dotted line provides a linear approximation of this proposed method’s performance, serving as a simplified or theoretical model against which other methods’ performances are measured.

Figure 6 exhibit that the proposed method exceled in identifying spoofing attacks by maintaining a superior ‘Message Reception Probability’ across varying distances between the transmitter and receiver. Projected method consistently achieved higher reception probabilities even as the distance increases, which highlights its robustness and effectiveness in real-world scenarios with fluctuating signal conditions. Maintaining high message reception probabilities regardless of distance was essential for reducing the system’s vulnerability to signal degradation or disruptions caused by spoofing attacks. This capability ensured that communication integrity remained intact. The proposed method’s ability to preserve connectivity under changing conditions has significantly enhanced the reliability and security of vehicular networks, where transmission distances often change due to vehicular mobility. Alongside high reception probabilities, the method also reduced communication latency, which was critical for real-time data transmission in VANETs. Lower latency enabled faster message exchanges between nodes, which was vital for timely decision-making processes in dissimilar applications. Moreover, in our experimental settings, the throughput remained consistently high, showing that the network can manage a substantial volume of data packets without significant loss or delay, even when facing the challenge of a spoofing attack. This high throughput was essential for maintaining the flow of crucial information such as positional data and safety alerts, which are necessary for vehicular coordination and accident prevention. By combining high message reception probability, low latency, and high throughput, the proposed method not only effectively detected spoofing attacks but also maintained optimal network performance, rapid response times, and robust security.

Summary

Our experimental investigation conclusively demonstrated that the hybrid machine learning approach, integrating Autoencoders with LSTM networks has significantly enhanced the detection and mitigation of spoofing attacks by achieving an average accuracy of 94.98%, precision of 94.37%, and recall of 93.24%. These metrics underscore the robustness of our detection system in an emulated environment which reflects a broad spectrum of spoofing scenarios. The application of ABE further fortified the security framework, ensuring data integrity across vehicular communication channels. Our findings assert the critical role of advanced cryptographic and machine learning techniques in safeguarding intelligent transportation systems against emerging cyber threats. This study not only extends the current understanding of GNSS spoofing countermeasures but also sets a new benchmark for future research in vehicular network security, emphasizing the necessity for continuous adaptation and enhancement of security protocols to thwart sophisticated spoofing strategies effectively.

Conclusion

The presented research conveyed a thorough assessment of BeiDou signal spoofing in VANETs, addressing a critical security issue to modern vehicle communications. By integrating hybrid machine learning methods, specifically Autoencoders and LSTM networks, with the advanced cryptographic technique of ABE, the study proposed a robust framework that significantly enhanced the detection and mitigation of spoofing attacks. The proposed methodology focused on a dual approach: leveraging machine learning for dynamic threat detection and applying cryptographic security to ensure data integrity. Our experimental results confirmed the effectiveness of the strategies employed in an emulated environment, showing their promise for application in real-world settings. Nonetheless, the study acknowledges limitations such as variability in signal conditions and the controlled nature of these experiments. The findings highlight the critical need to improve security within vehicular networks and lay the groundwork for future research that should focus on refining projected methods. Future work should aim to expand the variety of test scenarios and tailor the model to oversee various spoofing attacks by facilitating better integration of these strategies into existing vehicular infrastructure and stirring towards a more secure and robust vehicular communication system.

Supplemental Information

Supplemental Information 1 Code

Additional Information and Declarations

Competing Interests

Author Contributions

Data Availability

The authors declare there are no competing interests.

Usman Tariq conceived and designed the experiments, performed the experiments, analyzed the data, performed the computation work, prepared figures and/or tables, authored or reviewed drafts of the article, and approved the final draft.

The following information was supplied regarding data availability:

The data supporting the findings of this study are available at Zenodo: Tariq, U. (2024). BDS Spo. Ident., Mang., and To.: Lah. & Khj. (Version v1) [Data set]. UT_PSAU. https://doi.org/10.5281/zenodo.12570444.

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
