# Peer review of "Intelligent algorithmic framework for detection and mitigation of BeiDou spoofing attacks in vehicular ad hoc networks (VANETs)"

_PeerJ Computer Science, doi:10.7717/peerj-cs.2419_

## Round 0.1 · original submission · Major Revisions

Two reviewers have commented on this submission. They seem to agree that this manuscript may be publishable after a major revision. The first reviewer raises a number of important points regarding the novelty of this work, and in particular asks a comparison of the performance of the new method with the state of the art for the problem.

·

Basic reporting

Manuscript ID Submission ID 102709v1
This paper is related to reviewing the manuscript titled " Intelligent algorithmic framework for detection and mitigation of BeiDou spoofing attacks in VANETs"
This research examines the critical issue of spoofing in the BeiDou Navigation Satellite System (BDS) within Vehicular Ad-hoc Networks (VANETs), offering advanced strategies for detection, tolerance, and management to enhance vehicular communication security. With the growing reliance on BDS for accurate vehicle positioning, spoofing presents significant risks to vehicular safety and traffic management. The authors use a hybrid machine learning approach, integrating Autoencoders and Long Short-Term Memory (LSTM) networks, along with the advanced cryptographic method ‘Attribute-Based Encryption (ABE)’, to create a robust anti-spoofing framework.

Experimental design

Firstly, Although the proposed study is successful in terms of organization, presentation, content and results, major revision given in the following items need to be performed.
1) Provide the major numerical findings and conclusions of the study in the summary section.
2) The mathematical model of proposed LSTM Model must be validated. Why is the recommended model a hybrid model? How is it strikingly different from others?
3) Standard LSTM model equations are given in Equations 10-13. The innovation and contribution of the proposed deep network-based model must be given by the proposed model.
4) The proposed method lacks any basis regarding VANET energy consumption.
5) Increase the resolution of figures.
6) Several operations were carried out on the VANET network with the method of attack detection and mitigation. However, it seems that performance analyzes that are widely used in attack detection and prevention studies, such as error checking, collision rate, efficiency, and data loss rate, are not given in the experimental part.
7) In addition, the proposed model should be compared with new methods.

Validity of the findings

As above

Additional comments

My decision is major revision. I do not see any harm in publishing the manuscript once the above revisions are made.

Reviewer 2 ·

Basic reporting

1.The abstract and conclusion of this article need to be rewritten because the author did not clearly state what solutions or methods the article proposed, or what problems it solved.
2. This article should conduct a complexity analysis and discussion on the proposed algorithm.
3. This article lacks a theoretical feasibility analysis of the proposed solution.

Experimental design

The paper should provide a more detailed analysis of the reasons behind the experimental results.

Validity of the findings

no comment

Additional comments

1.The abstract and conclusion of this article need to be rewritten because the author did not clearly state what solutions or methods the article proposed, or what problems it solved.
2. This article should conduct a complexity analysis and discussion on the proposed algorithm.
3. This article lacks a theoretical feasibility analysis of the proposed solution.

---

## Round 0.2 · accepted · Accept

The reviewers have no further comments, and they both recommend that the manuscript be accepted as is.

·

Basic reporting

The paper can be accepted

Experimental design

The paper can be accepted

Validity of the findings

The paper can be accepted

Additional comments

The paper can be accepted

Reviewer 2 ·

Basic reporting

no comment

Experimental design

no comment

Validity of the findings

no comment

Additional comments

no comment